Seasonal and predator-prey effects on circadian activity of free-ranging mammals revealed by camera traps

http://orcid.org/0000-0002-1763-8970 Caravaggi Anthony 1 2 ar.caravaggi@gmail.com
Gatta Maria 3
Vallely Marie-Claire 1 4
Hogg Kayleigh 1
Freeman Marianne 1
Fadaei Erfan 1 5
Dick Jaimie T.A. 1 5
Montgomery W. Ian 1 5
Reid Neil 1 5
Tosh David G. 1 6
1 School of Biological Sciences, Queen’s University Belfast , UK
2 School of Biological Earth and Environmental Sciences, University College Cork , Cork , Ireland
3 School or Animal, Plant and Environmental Sciences, University of Witwatersrand , South Africa
4 Northern Ireland Environment Agency , UK
5 Institute of Global Food Security (IGFS), Queen’s University Belfast , UK
6 National Museums Northern Ireland , UK
Forbes Valery
Electronic publication date: 2018 Nov 21
Publication date: 2018
Volume: 6
Electronic Location ID: e5827
Received 2018 Mar 21; Accepted 2018 Sep 25
Copyright: © 2018 Caravaggi et al.
Copyright year: 2018
Copyright holder: Caravaggi et al.
License: This is an open access article distributed under the terms of the Creative Commons Attribution License, which permits unrestricted use, distribution, reproduction and adaptation in any medium and for any purpose provided that it is properly attributed. For attribution, the original author(s), title, publication source (PeerJ) and either DOI or URL of the article must be cited.
License URL: https://creativecommons.org/licenses/by/4.0/

Keywords: Temporal co-occurrence, Mammal species, Circadian activity, Camera traps, Citizen science, Seasonality, Wildlife

Funding: Northern Ireland Challenge Fund Northern Ireland Environment Agency’s Natural Heritage Research Partnership Natural Heritage Research Partnership (NHRP) Northern Ireland Environment Agency (NIEA) and Quercus Queen’s University Belfast (QUB) People’s Trust for Endangered Species (PTES) Department of Environment, Agriculture and Rural Affairs (DAERA) or Department for the Economy (DfE) Both squirrel studies were funded under the Northern Ireland Challenge Fund and supported by the Northern Ireland Environment Agency’s Natural Heritage Research Partnership. Hare surveys were commissioned and funded by the Natural Heritage Research Partnership (NHRP), between the Northern Ireland Environment Agency (NIEA) and Quercus, Queen’s University Belfast (QUB) with part-funding from the People’s Trust for Endangered Species (PTES). Deer surveys were part of Department of Environment, Agriculture and Rural Affairs (DAERA) or Department for the Economy (DfE) funded PhD studentships. The funders had no role in study design, data collection and analysis, decision to publish, or preparation of the manuscript.

==============================
Endogenous circadian and seasonal activity patterns are adapted to facilitate effective utilisation of environmental resources. Activity patterns are shaped by physiological constraints, evolutionary history, circadian and seasonal changes and may be influenced by other factors, including ecological competition and interspecific interactions. Remote-sensing camera traps allow the collection of species presence data throughout the 24 h period and for almost indefinite lengths of time. Here, we collate data from 10 separate camera trap surveys in order to describe circadian and seasonal activity patterns of 10 mammal species, and, in particular, to evaluate interspecific (dis)associations of five predator-prey pairs. We recorded 8,761 independent detections throughout Northern Ireland. Badgers, foxes, pine martens and wood mice were nocturnal; European and Irish hares and European rabbits were crepuscular; fallow deer and grey and red squirrels were diurnal. All species exhibited significant seasonal variation in activity relative to the timing of sunrise/sunset. Foxes in particular were more crepuscular from spring to autumn and hares more diurnal. Lagged regression analyses of predator-prey activity patterns between foxes and prey (hares, rabbits and wood mice), and pine marten and prey (squirrel and wood mice) revealed significant annual and seasonal cross-correlations. We found synchronised activity patterns between foxes and hares, rabbits and wood mice and pine marten and wood mice, and asynchrony between squirrels and pine martens. Here, we provide fundamental ecological data on endemic, invasive, pest and commercially valuable species in Ireland, as well as those of conservation importance and those that could harbour diseases of economic and/or zoonotic relevance. Our data will be valuable in informing the development of appropriate species-specific methodologies and processes and associated policies.

Introduction

Animal activity patterns are influenced by a variety of environmental pressures, including food availability (Larivière, Huot & Samson, 1994; Pereira, 2010), foraging efficiency (Lode, 1995; Prugh & Golden, 2014), predator/prey activity (Fenn & Macdonald, 1995; Middleton et al., 2013), human disturbance (Van Doormaal et al., 2015; Wang, Allen & Wilmers, 2015), mate availability and activity (Thompson et al., 1989; Halle & Stenseth, 2000), and ecological competition (Rychlik, 2005; Monterroso, Alves & Ferreras, 2014). Circadian (i.e. recurring every 24 h) and seasonal patterns of activity are adaptive behavioural traits which allow species to effectively exploit their environment and the resources contained therein (Hetem et al., 2012; Phillips et al., 2013). Mammals exhibit a great diversity and flexibility in their activity patterns. A recent study of 4,477 mammal species classified 69% as nocturnal (i.e. night-active), 20% diurnal (i.e. day-active), 8.5% cathemeral (i.e. active throughout the 24 h cycle) and 2.5% crepuscular (i.e. dawn- and/or dusk-active, e.g. lesser mouse deer; Bennie et al., 2014).

Activity patterns evolved in response to cyclical changes in the environment that encouraged organisms to respond on a physiological and behavioural basis (Daan, 1981; Kronfeld-Schor & Dayan, 2003; Roll, Dayan & Kronfeld-Schor, 2006; Bennie et al., 2014). Activity patterns are frequently related to daily oscillation in illumination (e.g. changes in sunrise/sunset; Halle & Stenseth, 2000) and, hence, the time(s) of the day during which species are active may vary according to season (i.e. spring, summer, autumn and winter). Indeed, it has been suggested that photic cues are the dominant factor underlying behavioural rhythmicity and that a species’ potential to adapt to non-photic cues (e.g. ecological competitors, predators) may be constrained such that responses are manifest within the normal active period rather than as a shift to a different rhythm (Kronfeld-Schor & Dayan, 2003). The capacity for adaptive behavioural plasticity, while limited in some species (Kronfeld-Schor & Dayan, 2003), is demonstrated in others by observations of intraspecific variation of activity patterns (Ashby, 1972; Hertel et al., 2016) that can result in temporal niche switching (Fenn & Macdonald, 1995; Ensing et al., 2014). For example, Ensing et al. (2014) found that red deer (Cervus elaphus) were mostly diurnal in Canada, while conspecifics in the Netherlands were mostly nocturnal. The difference in activity patterns was attributed to higher levels of human disturbance and a lack of natural predators in the Netherlands.

The island of Ireland has a depauperate terrestrial mammalian community due to its prolonged isolation since the last Glacial Maximum (Montgomery et al., 2014). The behavioural ecologies of mammals in Ireland are almost entirely unknown and assumptions of behavioural equivalence between Ireland and locations elsewhere in the species’ range may not apply given differences in land use, human activity and ecosystem composition. Quantifying fundamental ecological parameters such as activity patterns has direct relevance to the management of endemic (e.g. Irish hare, Lepus timidus hibernicus, Bell 1837; Reid & Montgomery, 2007), invasive (e.g. European brown hares, L. europaeus, Pallas 1778; Caravaggi, Montgomery & Reid, 2015), pest (e.g. red foxes, Vulpes vulpes, Linnaeus 1758; Baker & Harris, 2006) and commercially valuable (e.g. deer; Carden et al., 2011) species, as well as those that could harbour diseases of economic (e.g. badgers, Meles meles, Linnaeus 1758; Griffin et al., 2005) and/or zoonotic relevance. The relative dearth of mammalian herbivores and their predators in Ireland also makes the island an ideal study system in which to investigate predator-prey relationships. However, recording and quantifying daily activity patterns of wild, free-ranging mammals presents significant challenges, including overcoming the observer effect whereby the presence of an observer influences the behaviour of the subject (Stewart, Ellwood & Macdonald, 1997), and collecting sufficient data to address scientific and conservation questions (sensu Cagnacci et al., 2010). A number of methodological techniques have been used to overcome such challenges such as radio-tracking, GPS collars and live trapping, each with varying degrees of success (Bridges & Noss, 2011). Radio-tracking has inherent limitations, including periodic (i.e. non-constant) sampling (Lovari, Valier & Lucchi, 1994) and the application of considerable survey effort (Palomares & Delibes, 1991; Reid, McDonald & Montgomery, 2010). Furthermore, they may result in small sample sizes (Bridges & Noss, 2011), capture a limited proportion of the population (Sadlier et al., 2004), and may be subject to signal-based error and/or omission (Cagnacci et al., 2010) or alter the behaviour of tagged animals (Wilson et al., 2011). GPS collars have similar constraints to radio-tracking, particularly with regards to sample sizes and potential signal issues. Moreover, inferences made from GPS collar data can lead to misleading results and much depends on the frequency with which location fixes are obtained (Merrill & Mech, 2003). Live trapping has been used to investigate activity patterns of small mammals, where each successful capture (i.e. the presence of an animal in a trap) is taken as indicating activity (e.g. Elton et al., 1931; Bradley, 1967; Hoogenboom et al., 1984). However, live trapping requires considerable time and effort, is relatively inefficient, may have implications with regards to animal welfare (Torre, Guixe & Sort, 2010), and is subject to species- and trap-specific variations in capture probability (Leso & Kropil, 2010).

Remote-sensing camera traps (i.e. remotely activated cameras that are activated via motion or infra-red sensors or light beams; Swann et al., 2004) are increasingly popular in conservation and ecological studies (Kucera & Barrett, 2011) as they are non-invasive and are subject to continuing technological improvements and decreasing costs (Tobler et al., 2008a). They have been used in studies investigating population densities (Trolle & Kéry, 2003; Karanth et al., 2006; Caravaggi et al., 2016), behaviour (Maffei et al., 2005), ecosystem biodiversity (Silveira, Jácomo & Diniz-Filho, 2003; Tobler et al., 2008b), and site occupancy of rare or cryptic species (Linkie et al., 2007). Camera traps afford researchers the means to conduct long-term surveys while minimising in situ survey effort and disturbance of the focal species (Kays & Slauson, 2008). As such, data derived from camera trap surveys of common species and/or conducted at high camera densities are well suited to investigations of wildlife activity patterns (Di Cerbo & Biancardi, 2013; Carbajal-Borges, Godínez-Gómez & Mendoza, 2014; Allen, Peterson & Krofel, 2018). The size and scope of camera trap surveys are limited only by the cost of equipment and personnel to conduct fieldwork and review and analyse data, while the length of time cameras can be left in situ is restricted by available memory, battery life and the possibility of mechanical failure. There may be a trade-off between the proximity and angle of cameras with regards to targets, and the likelihood of detecting and identifying species of varying size (Hofmeester, Rowcliffe & Jansen, 2017). Downward-facing cameras, for example, are more efficient at detecting small mammals (De Bondi et al., 2010). However, species identification is difficult where similar species occur in sympatry (Claridge, Paull & Barry, 2010; Meek, Vernes & Falzon, 2013; Oliveira-Santos et al., 2010). This is often particularly true of small mammals (Claridge, Paull & Barry, 2010; Meek & Vernes, 2016). However, Ireland is home to six species of rodent and two species of shrew, few of which occur in sympatry and all of which are uniquely identifiable.

Here, we describe the first study into temporal activity patterns of 10 mammal species found in Ireland, from 14 g (wood mouse, Apodemus sylvaticus, Linnaeus 1758) to 60 kg (fallow deer, Dama dama, Linnaeus 1758), captured via camera traps. In addition, we investigate interspecific relationships, specifically whether predator-prey activity patterns demonstrate (a)synchrony, demonstrating temporal (dis)association between ecologically-linked species. We hypothesised that predator-prey pairs would exhibit non-random, interrelated distribution of activity throughout the diel cycle. This distribution would either display a considerable overlap between predators and prey (i.e. predators are attracted to prey), or asynchrony between the species (i.e. temporal avoidance of predators by prey).

Materials and Methods

We collated data from 10 camera trap studies conducted in Northern Ireland, where land use is predominantly agricultural (75%) and forest cover, mixed and deciduous woodland and coniferous forest plantations, is 8% of land area (Department of Agriculture, Environment and Rural Affairs, 2018; Forestry Comission, 2018). The focal species of the 10 studies were fallow deer (Dama dama, Linnaeus 1758; n = 6), Eurasian red squirrel (Sciurus vulgaris, Linnaeus 1758) and Eastern North American grey squirrel (S. carolinensis, Gmelin 1788; n = 2) and European brown hare and Irish mountain hare (n = 2). However, as is common for camera trap studies, non-target species were also detected. Therefore, in addition to the five focal species, we present additional information on a further five species: (i) European badger; (ii) European rabbit (Oryctolagus cuniculus, Linnaeus 1758); (iii) wood mouse; (iv) pine marten (Martes martes, Linneus 1758); and (v) red fox (hereafter ‘fox’). The data presented here resulted from a total of 1,164 camera deployments at 431 locations (defined herein as broad study areas, rather than individual camera placements; Fig. 1). Deer surveys were conducted from June 2013 to November 2016, squirrel surveys from January to March 2014 and January to May 2015 and hare surveys from April 2013 to August 2015, non-inclusive. Constituent surveys were independent thus methodologies were not standardised. There was no evidence of intrageneric variation in the activity patterns of hare (Fig. S1) and squirrel (Fig. S2) species, and, hence, both were grouped (i.e. ‘hares’ and ‘squirrels’) for the purposes of the current study.

Figure 1 Locations of sites used in camera trap wildlife studies in Northern Ireland from 2013 to 2016.

For species-specific maps, see Fig. S3.

Species surveys

We included data from six independent deer studies (D1—6) in our analyses. D1 was conducted over 15 1 km2 squares with an average of 10 cameras per km2 and five additional one km2 squares set at a higher density of 20 camera traps per km2. In total 38 camera trap units were randomly deployed over 255 individual camera trap placements (Table 1) using a combination of Bushnell Trophy Cam (119467), Bushnell Trophy Cam HD (119477), Reconyx (HC600) and Scoutguard Camera (SG560P-8M)—the number of each model used differed between sites. Camera traps were set at a height of 30 cm, perpendicular to the ground. Cameras were set to capture the maximum photographs per trigger (3–10 photographs depending on camera model) and no delay between triggers. Cameras were left for 14 days before being collected and relocated. D2–D4 surveyed smaller areas of 0.02, 0.04 and 0.05 km2 using 10 Bushnell Trophy Cam HD (119677) at each site. Each camera was set at a height of 40 cm from the ground and set to capture bursts of three still pictures and a 60 s video per trigger, with a delay of 1 s between triggers. Cameras were left in situ for 7 days. D5–D6 were focussed on areas of 0.05 and 0.02 km2, respectively and used Bushnell Trophy Cam HDs (119477, 119577, 119676, 119677). Cameras were set at a height of 40 cm and set to capture either three still pictures or a 30 s video, depending on the camera model, with a 1 s delay between triggers. Cameras were deployed for 7 days.

Table 1 Number of remote-sensing camera traps deployed (i.e. ‘camera locations’) by mammal studies carried out in Northern Ireland between 2013 and 2016.

Focal species	Year	Total survey area (km2)	Active cameras	Deployments	Total camera locations	
Deer (D1)	2013–2014	20.00	38	23	255	
Deer (D2)	2015	0.05	10	4	40	
Deer (D3)	2015	0.02	10	2	20	
Deer (D4)	2015	0.04	10	2	20	
Deer (D5)	2015	0.05	17	1	17	
Deer (D6)	2015	0.02	21	1	21	
Hares	2013–2014	17.00	20	17	340	
Hares	2015	6.00	12	6	72	
Squirrels (S1)	2014	n/a	16	63	63	
Squirrels (S2)	2015	n/a	65	314	314	
Note:

Contributory studies were independent, thus methodologies were not standardised; study locations (size, shape) and camera array densities varied considerably. For more information, see the main Methods section. Squirrel surveys were focussed on presence and did not attempt to quantify the effective survey area of all camera placements.

Hare surveys were conducted over two independent studies (Table 1) across a total of 23 1 km2 squares. Each square contained 20 randomly placed Bushnell Trophy Cam HD (119477) camera traps that were positioned on vertical aspects of linear features (i.e. trees in hedgerows, fence posts), to a total of 412 camera locations (Table 1). Cameras were set at a height of 30 cm from the ground, at a 45–90° angle away from the linear feature, with a 10–15° downward tilt. Cameras were set to record video for a period of 60 s with a 60 s delay between triggers. Cameras were deployed for 7 days. The use of video footage allowed the detection of closely-associated conspecifics (see Caravaggi et al., 2016 for more on this study).

Data from two arboreal squirrel surveys are presented in this study. The first survey (S1) was undertaken in 2014, within 63 forested areas >5 ha in size within Co. Fermanagh. A total of 16 Bushnell Trophy Cameras (119438) were deployed by seven volunteers (‘citizen scientists’) and one scientist during a 3-month period (Table 1). Cameras were attached to trees at a height of three to four m and positioned up to four m distant from and opposite a wooden squirrel feeder (Northumbrian Wildlife Trust design) baited with peanuts and sunflower seeds. Cameras were left in situ for a minimum of seven and maximum of 24 days (median = 8). The second survey (S2) was conducted in 314 forested areas >5 ha in size across Northern Ireland by 70 citizen scientists and one scientist during a 5-month period in 2015. A total of 65 Bushnell Trophy Cameras (16 × 119438; 12 × 119577; 37 × 119676) were deployed during this time for a minimum of six and maximum of 33 days (median = 7). Cameras were deployed at head height (1.5–2 m) on a tree opposite either a wooden (as in 2014) or metal squirrel feeder (CJ Wildlife Product code 12335). In both years, cameras were set to take three photos per burst with an interval of 1–20 s between triggers. In these surveys all citizen scientists were trained in the use of camera traps by the scientist (DT) overseeing the research.

In deer and hare surveys, cameras were placed according to ni randomly generated points within each focal area and a clear field of view was ensured by clearing prominent vegetation where appropriate. The density of deployed camera trap arrays means that it was possible that the same individual would be captured more than once during the sampling period. Similarly, cameras were pseudo-randomly placed during squirrel surveys according to suitably paired trees (one for the camera, one for the feeder) of which there were an abundance in all locations. Recaptures were highly likely given the use of baited stations. Surveys with >1 camera trap model did not systematically deploy models predictably within and across arrays. In all surveys, cameras were equipped with eight GB HDSD cards, secured with Python security cables, motion detectors were set to medium sensitivity and each capture was stamped with the date and time.

Activity analysis

We assumed that images were independent when separated by 1 h (Cusack et al., 2015) and that temporal detection frequency was a true reflection of circadian activity patterns of the focal species. Animals can only be detected on camera traps when active, specifically, when in motion. The number of detections per hour therefore reflects the level of activity across the circadian period. Prior to analysis, data for individual species were grouped into 1 h time intervals, beginning at the hour mark (e.g. 11:00–11:59). In cases where a group of individuals of the same species (e.g. fallow deer) was detected in one image, a single event was recorded. Detection frequencies were normalised to ease plot interpretation using the formula zi=xi−xminxmax−xmin, where zi = normalised detection frequency at the ith interval, and x = (x1…, xn).

Variations in day length throughout the year has direct consequences for wildlife (Nouvellet et al., 2012). For example, an ostensibly nocturnal species may be more likely to remain active into daylight hours during the summer months due to shortening nights in order to meet its energetic requirements (Schai-Braun, Rödel & Hackländer, 2012). To investigate intra-annual variation in activity relative to sunrise/sunset (i.e. whether nocturnal/diurnal activity differed between seasons; solar cycle historical data obtained from the HM Nautical Almanac Office, 2016), data were grouped according to season: spring (March—May); summer (June—August); autumn (September—November); and winter (December—February). Detections between 00:00 and 11:59 were offset relative to sunrise; detections between 12:00 and 23:59 were offset relative to sunset. All daytime offsets were converted to positive integers, night-time to negative. For example, a detection timed at 22:10, with sunset at 20:00, would have an offset value of −2 h and 10 min, indicating nocturnal activity. We investigated intraspecific differences in seasonal offsets via one-way Analysis of Variance with post hoc Tukey tests where the dependent variable was the sunrise/sunset offset, and season was the explanatory variable.

We defined the diurnal period as the time between 1 h after sunrise and 1 h before sunset, and the nocturnal period as the time between 1 h before sunrise and 1 h after sunset. We defined the crepuscular periods, dawn and dusk, as the hour before and the hour after sunrise and sunset, respectively, (after Theuerkauf et al., 2003; Ross et al., 2013). Species activity patterns were classified according to the diel period with the most activity.

We used overlap metrics and lagged regression cross-correlation functions (CCFs) to examine annual and seasonal relationships between predator-prey pairs (i.e. autocorrelation), specifically: fox and hare; fox and rabbit; fox and wood mouse; pine marten and squirrel; and pine marten and wood mouse. These relationships were explored due to dietary studies indicating the potential for predator-prey dynamics within the 24 h period. Lagomorphs are primary prey items of the red fox in Ireland (Fairley, 1970). In Britain the field vole (Microtus agrestis, Linnaeus, 1761) is a secondary prey item for the red fox (Webbon et al., 2006) and primary item for the pine marten (Caryl et al., 2012) but this species does not occur in NI where the wood mouse is the most abundant small mammal species (J. P. Twining, 2018, unpublished data). Data used to examine predator prey relationships were restricted to locations where both species in each pair were detected. Sample CCFs facilitate the identification of lags in the x variable which may be predictive of y. Positive lag (h+) is the result of a correlation between xa+i and ya, where a = time. Conversely, negative lag (h−) is the result of a correlation between xa−i and ya. Significant correlations describe a non-random association between species detections at interval(s) hi. Lagged regressions were calculated using the CCF function in the core library of R (R Core Team, 2016). The CCF function, however, does not return quantified measures of significance. The significance of the correlation coefficient, r, therefore, was established by calculating the t value, where t=rn−21−r2 and where the critical t value (p = 0.05, 22 degrees of freedom, one-tailed) = 1.72. We calculated the degree of overlap between each species pair on an annual basis and for each season using the overlap package (Meredith & Ridout, 2017). Data were resampled 1,000 times per pair, per category, to generate 95% Confidence Intervals (CIs). We defined overlap <0.5 as low overlap, 0.5–0.75 as moderate overlap and > 0.75 as high overlap (Monterroso, Alves & Ferreras, 2014). All statistical analyses were carried out in R 3.4.3 (R Core Team, 2017; see Caravaggi et al., 2018, for data and code).

Results

A total of 8,761 independent detections of the 10 species were recorded across 324 camera days (i.e. 24-hour periods across all deployed cameras). Squirrel sightings (n = 2,870; Fig. 2G) comprised 33% of all records, rabbits 13% (n = 1,175; Fig. 2F), pine martens 11% (n = 966; Fig. 2E), badgers 11% (n = 947; Fig. 2A), wood mice 9% (n = 816; Fig. 2H), hares 9% (n = 751; Fig. 2D), foxes 7% (n = 645; Fig. 2D) and fallow deer 7% (n = 591; Fig. 2B; Table 2). Seasonal variations in the number of detections recorded reflected the time and duration of the constituent studies: 47% in spring, 24% in summer, 18% in autumn and 11% in winter (Table 2).

Figure 2 Camera trap images of 10 mammal species detected in Northern Ireland between 2013 and 2016.

(A) Badger, (B) fallow deer, (C) fox, (D) European hare, (E) Irish hare, (F) pine marten, (G) rabbit, (H) grey squirrel, (I) red squirrel and (J) wood mouse (circled). Images provided by A. Caravaggi A, D, E, J, K. Hogg and M. Freeman (B, C, G) and D.G. Tosh F, H, I.

Table 2 Total number of species detections during camera trap surveys in Northern Ireland from 2013 to 2016.

Species	Season		
Common name	Latin name	BM (kg)	Spring	Summer	Autumn	Winter	Σ	
Fallow deer	Dama dama	57.00	38	484	61	8	591	
Badger	Meles meles	11.00	618	225	36	68	947	
Fox	Vulpes vulpes	4.80	198	183	149	115	645	
Hare	Lepus sp.	3.46	301	339	105	6	751	
Rabbit	Oryctolagus cuniculus	1.59	492	417	238	28	1,175	
Pine marten	Martes martes	1.30	251	73	356	286	966	
Squirrel	Sciurus sp.	0.44	1,798	317	462	293	2,870	
Wood mouse	Apodemus sylvaticus	0.02	449	57	119	191	816	
Note:

Hare = Irish hare (Lepus timidus hibernicus) and European hare (L. europaeus); squirrel = grey squirrel (Sciurus carolinensis) and red squirrel (S. vulgaris). BM = body mass (Jones et al., 2009). Hare and squirrel body mass are given as a mean of the two detected species: European hare = 3.82 kg; Irish hare = 3.11 kg; grey squirrel = 0.55 kg; red squirrel = 0.33 kg. Species are ordered according to body mass.

Species-specific activity patterns

Foxes exhibited a nocturnal activity pattern, with some irregular diurnal activity. Nearly three-quarters of all fox activity (73%) occurred between 21:00 and 07:00 (Fig. 3A). Pine marten activity was nocturnal, with 70% of all detections occurring between 21:00 and 06:00 (Fig. 3B). Badgers were nocturnal with a unimodal pattern of activity; the number of detections increased rapidly after dusk and decreased rapidly around dawn. Fewer than 15% of all badger detections were recorded between 06:00 and 19:00, indicating little diurnal activity (Fig. 3C). Hare activity patterns were bimodal, demonstrating predominantly crepuscular behaviour with 71% of all activity occurring between 04:00 and 08:00 (47%) and 20:00–23:00 (24%; Fig. 3A). Rabbits were also crepuscular, with 35% and 32% of all detections occurring between 04:00 and 08:00 and 17:00–23:00, respectively (Fig. 3A). Squirrels were diurnal, being active from dawn to dusk with fewer than 5% of triggers occurring between 19:00 and 05:00 (Fig. 3B). Wood mice were nocturnal with 81% of activity occurring between 21:00 and 06:00 (Fig. 3B). Fallow deer were diurnal with 63% of detections occurring between 06:00 and 18:00 (Fig. 3C).

Figure 3 Circadian activity patterns of 10 mammal species detected during camera trap surveys in Northern Ireland from 2013 to 2016.

(A) Fox (Vulpes vulpes) hare (Irish hare, Lepus timidus hibernicus and European hare, L. europaeus; see Fig. S1) and rabbit (Oryctolagus cuniculus); (B) pine marten (Martes martes), wood mouse (Apodemus sylvaticus) and squirrel (grey squirrel, Sciurus californicus and red squirrel, Sciurus vulgaris; see Fig. S2); (C) badger (Meles meles) and fallow deer (Dama dama). Shaded areas represent night time.

There were significant differences in offsets across seasons in fox activity patterns (F33,641 = 23.36, p < 0.0001); winter was significantly different from all other seasons due to decreased activity during daylight and crepuscular periods (Fig. 4C). Significant differences were observed across and between all seasons (F3,962 = 86.28, p < 0.0001), except spring-autumn in pine marten activity patterns (Fig. 4B). Badger activity patterns exhibited significant differences in offsets across seasons (F3,943 = 32.54, p < 0.0001; Fig. 4C) except for winter-autumn and summer-spring. Seasonal offsets differed significantly in hare activity patterns (F3,747 = 19.33, p < 0.0001), specifically between spring and summer (p < 0.0001) and summer and autumn (p < 0.001; Fig. 4D); hares exhibited more diurnal activity both in spring and in autumn. There were significant differences in offsets across and between seasons in rabbit (F3,1171 = 12.93, p < 0.0001; Fig. 4F) and squirrel activity patterns (F3,2866 = 76.52, p < 0.0001; Fig. 4G). Offsets were significantly different across seasons in wood mouse activity patterns (F3,812 = 39.31, p < 0.0001), with diurnal activity increasing in summer (Fig. 4H). Fallow deer activity patterns exhibited significant differences in offsets across seasons (F3,587 = 16.7, p < 0.0001), specifically between spring and autumn (p < 0.0001) and summer and autumn (p > 0.0001; Fig.4B).

Figure 4 Time of detection relative to sunrise/sunset during spring, summer, autumn and winter for 10 mammal species observed during camera trap surveys in Northern Ireland between 2013 and 2016.

(A) Badger, (B) fallow deer, (C) fox, (D) hare (Irish hare and European hare), (E) pine marten, (F) rabbit, (G) squirrel (grey squirrel and red squirrel), and (H) wood mouse. The upper, unshaded area denotes daytime, the lower, shaded area denotes night. Dashed lines indicate mean annual offset. Boxes represent the mean ± Standard Deviation. Raincloud plots (Allen et al., 2018a, 2018b) represent the density and spread of all contributing data points.

Predator-prey relationships

There was evidence of correlative relationships between all predator-prey pairs, both annually and between seasons. Fox and hare annual activity patterns showed 73% overlap (CI [68–77%]; Table 3) and were significantly positively correlated with a peak at −2 h (peak lag = pl, hereafter; r = 0.663, t22 = 4.15, p < 0.0005; Table 4). The degree of overlap peaked in spring at 75% (CI [64–84%]; Table 3; pl = 1 h, r = −0.554, t22 = 3.12, p < 0.005; Table 4). There were significant correlations in the summer (Table 4). Annual activity patterns of foxes and rabbits overlapped by 80% (CI [75–83%]; Table 3) and were significantly correlated, with a peak at −1 h (pl = −1 h, r = 0.661, t22 = 4.13, p < 0.0005; Table 4). Overlap was greatest during spring at 89% (CI [86–98%]; Table 3; pl = 1 h, r = 0.701, t22 = 4.61, p < 0.0005; Table 4) and lowest during winter (51 CI [31–71%]; Table 3). Seasonal activity patterns between foxes and rabbits were positively correlated during spring and summer (Table 4), but there were no significant correlations evident during the rest of the year. Fox and wood mouse annual activity patterns overlapped by 81% (CI [75–87%]; Table 3). Overlap was high in all seasons but was greatest in summer with at 85% (CI [81–92%]; Table 3). Annual activity was significantly correlated (pl = 1 h, r = 0.754, t22 = 5.39, p ≤ 0.0001; Table 4), with similar peaks in cross-correlation coefficients in spring (Table 4) and autumn, with greatest correlation occurring in summer (pl = 0 h, r = 0.761, t22 = 5.50, p ≤ 0.0001; Fig. 4; Table 4).

Table 3 Annual and seasonal overlap (%, with 95% Confidence Intervals (CIs)) in the activity patterns of five predator-prey pairs.

Species		Season	
Predator	Prey	Annual	Spring	Summer	Autumn	Winter	
Fox	Hare	73 (68–77)	75 (64–84)	67 (60–73)	48 (31–52)	–	
Fox	Rabbit	80 (75–83)	89 (86–98)	78 (71–85)	52 (38–55)	51 (31–71)	
Fox	Wood mouse	81 (75–87)	78 (67–89)	85 (81–92)	69 (57–74)	68 (65–98)	
Marten	Squirrel	40 (33–41)	28 (16–30)	54 (41–60)	40 (32–41)	5 (0–5)	
Marten	Wood mouse	71 (64–74)	69 (57–82)	71 (56–82)	63 (52–66)	77 (66–93)	
Note:

Animals were detected during camera trap surveys in Northern Ireland between 2013 and 2016. Hare = Irish hare (Lepus timidus hibernicus) and European hare (L. europaeus); squirrel = grey squirrel (Sciurus carolinensis) and red squirrel (S. vulgaris). Few hares were detected during winter. Activity data were resampled 1,000 times per pair, per category, to generate CIs. For annual overlap plots, see Fig. S4.

Table 4 Temporal (dis)associations between activity patterns of five predator-prey pairs.

			Lag (hrs)				
Predator	Prey	Season	From	To	Peak lag	t	r	
Fox	Hare	Annual	−4	0	−2	4.15	0.663**	
Fox	Hare	Spring	−2	3	1	3.12	−0.554*	
Fox	Hare	Summer	−5	0	−3	3.54	0.602**	
Fox	Hare	Autumn	−3	−1	−3	2.08	0.405*	
Fox	Rabbit	Annual	−10	−8	−9	2.13	0.413*	
Fox	Rabbit	Annual	−3	1	−1	4.13	0.661**	
Fox	Rabbit	Annual	8	11	10	2.10	−0.409*	
Fox	Rabbit	Spring	−1	2	1	4.61	0.701***	
Fox	Rabbit	Spring	–	–	11	2.14	−0.415*	
Fox	Rabbit	Summer	−12	−9	−4	2.52	−0.473*	
Fox	Rabbit	Summer	−4	1	−1	4.06	0.654**	
Fox	Mouse	Annual	−12	−10	−11	2.56	−0.480*	
Fox	Wood mouse	Annual	−2	3	1	5.39	0.754***	
Fox	Wood mouse	Spring	−2	3	1	3.26	0.570*	
Fox	Wood mouse	Summer	−10	−9	−10	2.09	−0.407*	
Fox	Wood mouse	Summer	−1	1	0	5.50	0.761***	
Fox	Wood mouse	Autumn	0	2	2	3.04	0.544*	
Pine marten	Squirrel	Annual	−12	−7	−9	2.15	0.416*	
Pine marten	Squirrel	Annual	−1	4	1	3.72	−0.621**	
Pine marten	Squirrel	Spring	−10	−4	−6	2.50	0.470*	
Pine marten	Squirrel	Spring	0	5	1	3.76	−0.625**	
Pine marten	Squirrel	Summer	−4	−1	−2	2.79	−0.512*	
Pine marten	Squirrel	Summer	–	–	8	2.46	0.464*	
Pine marten	Squirrel	Winter	−9	−7	−8	2.79	0.511*	
Pine marten	Squirrel	Winter	−2	3	−1	4.18	−0.665**	
Pine marten	Squirrel	Winter	9	11	10	3.11	0.553*	
Pine marten	Wood mouse	Annual	−1	1	0	2.98	0.536*	
Pine marten	Wood mouse	Spring	7	11	10	2.29	−0.440*	
Pine marten	Wood mouse	Summer	3	−2	−2	2.24	0.431*	
Pine marten	Wood mouse	Summer	–	–	8	2.46	0.464*	
Pine marten	Wood mouse	Winter	2	4	3	2.73	−0.503*	
Pine marten	Wood mouse	Winter	6	7	6	2.05	0.400*	
Note:

Lag range and peak lag were calculated using cross-correlation functions (CCFs). t = t-value, where the critical value (p = 0.05, df = 22) = 1.72. r = correlation coefficient. Positive values indicate that detections of predators preceded/succeeded those of prey species. Negative values indicate the opposite. Statistical significance is indicated by asterisks, where, * ≤ 0.05; ** ≤ 0.001; *** ≤ 0.0001.

Pine marten and wood mouse annual activity patterns were correlated with a 71% (CI [64–74%]) overlap (pl = 0 h, r = 0.536, t22 = 2.98, p ≤ 0.05; Table 4). Overlap was greater than 50% for all seasons with a peak of 77% (CI [66–93%]) occurring in winter months (Table 3). Activity patterns were significantly correlated in spring (pl = 10 h, r = −0.44, t22 = 2.29, p ≤ 0.05; Table 4) and winter (pl = 3 h, r = −0.503, t22 = 2.73, p ≤ 0.05). Pine marten and squirrel annual patterns overlapped by 40% (CI [33–41%]; Table 3) and their activity was significantly correlated (pl = 2 h, r = −0.621, t22 = 3.72, p < 0.001; Table 4). Seasonal overlap peaked during summer at 54% (CI [41–60%]) but was almost entirely absent during winter (Table 3). Significant correlations between seasonal activity patterns were observed in all seasons except during autumn: spring, pl = 1 hr, (r = −0.625, t22 = 3.76, p > 0.001); summer, pl = −2 h (r = −0.512, t22 = 2.79, p < 0.001); and winter, pl = −1 h (r = −0.665, t22 = 4.18, p < 0.0005; Table 4).

Discussion

This study describes the first investigation into the activity patterns of a range of mammal species on the island of Ireland, adding to information gathered from across the species’ ranges and to those for which little is recorded. The study also contributes to the growing body of literature that uses remote camera traps to infer activity (Bridges & Noss, 2011; Monterroso, Alves & Ferreras, 2013; Ross et al., 2013; Carbajal-Borges, Godínez-Gómez & Mendoza, 2014; Allen, Peterson & Krofel, 2018) and reveals that our focal species exhibit differences in activity relative to sunrise/sunset and throughout the year. In some cases, activity changed markedly. For example, pine martens and rabbits were more diurnal in the summer while during the rest of the year they were nocturnal and cathemeral, respectively. Similarly, predator-prey relationships were also found to vary throughout the year. The temporal overlap between the activity patterns of foxes, lagomorphs (rabbits and hares) and wood mice, were high (>0.75) during the spring and summer and declined during autumn and winter. It is important to acknowledge that a wide variety of factors affect activity patterns and, while our data show positive correlations in activity between some predator and prey species that could be indicative of a causal relationship, we do not attempt to describe observed patterns as being driven by any specific stimulus.

Foxes were nocturnal with their activity increasing during dusk and decreasing at dawn, although there was also some activity during daylight. The bimodal activity pattern suggested by previous studies (Reynolds & Tapper, 1995) was not evident. Rural fox activity may be influenced by anthropogenic disturbance; diurnal activity is more common where disturbance is low (Díaz-Ruiz et al., 2016). In the present study, however, diurnal activity may have been facilitated, not by a lack of disturbance, but by the timing of the disturbance. Foxes are subject to nocturnal lethal control (i.e. shooting, facilitated by high-powered spot-lamps and other methods) across Northern Ireland, such that nocturnal disturbance by hunting parties is likely to be considerably greater than that which occurs during the day, albeit periodically. However, nocturnal control, while frequent, is regionally variable and intermittent. Furthermore, the most abundant and commonly taken prey animals are nocturnal (e.g. small rodents) or crepuscular (e.g. lagomorphs). Irregularities in the activity patterns described herein may therefore be indicative of the true activity signal along with occasional temporal displacement.

Hares were active throughout the night, with peaks of activity occurring between 20:00 and 23:00, and 04:00 and 08:00. Diurnal activity was most commonly recorded in the summer months, when nights are shortest (sensu Flux & Angermann, 1990; Holley, 1992; Langbein et al., 1999). This study is the first to quantify activity patterns of the Irish hare, which exhibited the same bimodal crepuscular-nocturnal behaviour as the European hare. The Irish hare is a subspecies of mountain hare that is endemic to Ireland while the European hare is almost exclusively found in Mid-Ulster, likely having been introduced in the 1970s (Caravaggi, Montgomery & Reid, 2015). Previous studies have demonstrated that the species are ecologically similar (Reid, 2011; Caravaggi, Montgomery & Reid, 2015), occur in sympatry (Caravaggi et al., 2016) and exhibit a high degree of bidirectional hybridisation (Prodohl et al., 2013). The similarity in activity patterns described herein adds additional support to the suggestion that strong interspecific competition is likely where they occur in sympatry where resources are limiting. The activity patterns of rabbits were similar to those in the Mediterranean region where rabbits were active throughout the day, but most activity occurred in the early morning and late afternoon (Monterroso, Alves & Ferreras, 2013). However, while our data suggest a similar bimodal pattern, the second, evening peak was weak in our dataset.

Overall, pine marten were nocturnal with activity becoming more diurnal between spring and early autumn, similar to patterns reported elsewhere in the species’ range (Monterroso, Alves & Ferreras, 2013; Zalewski, 2007; Zielinski, Spencer & Barrett, 1983). Previous studies have suggested that pine marten activity patterns may be linked to those of prey species (Zielinski, Spencer & Barrett, 1983). The relationship between predator and prey is highlighted by the suggestion that pine martens may play a role in the control of the invasive grey squirrel, to the benefit of the native red (Sheehy & Lawton, 2014). Pine marten activity in this study overlapped considerably with that of the wood mouse in summer and winter. Studies indicate the importance of wood mice to pine marten in Ireland in the absence of field voles (Lynch & McCann, 2007; O’Meara et al., 2013; J. P. Twining, 2018, unpublished data) and seasonal comparisons, in particular, show that wood mice are a major food source in all seasons but autumn (J. P. Twining, 2018, unpublished data) which is reflected in the correlation of activity reported here.

Squirrel activity peaked several hours after dawn and ceased before sunset. The species were almost exclusively diurnal between seasons, with detections occurring throughout the day (Tonkin, 1983; Gurnell & Hare, 2008). There was some evidence of variation of circadian patterns between seasons, with detections suggesting a bimodal pattern in the summer, and a unimodal pattern in the winter. The bimodularity observed in summer was caused by low activity around midday, possibly in response to increased temperatures (Tonkin, 1983; Zub et al., 2013). Similar seasonal variation in activity has been observed in previous studies (Thompson, 1977; Tonkin, 1983; Gurnell & Hare, 2008). Sunrise/set offsets revealed some crepuscular/diurnal activity, particularly during autumn and winter, behaviour that has not been reported previously (Tonkin, 1983; Wauters & Dhondt, 1987; Wauters, Swinnen & Dhondt, 1992). Foraging animals balance the risk of predation against the benefits of energy gains (Dammhahn & Almeling, 2012) and the acuity of squirrel eyesight is sub-optimal in low-light conditions (i.e. dawn/dusk; Jacobs, Birch & Blakeslee, 1982). Crepuscular activity may, therefore, be a response to local, diurnal predator activity, thus rendering an apparently suboptimal strategy contextually advantageous (Dammhahn & Almeling, 2012). The drivers of the observed crepuscular behaviour are, however, unknown and are worthy of further study.

Wood mice were nocturnal, although there was an increase in crepuscular and diurnal activity recorded during summer when shorter nights may provide insufficient foraging time for lactating female wood mice to meet their daily energetic requirements. Juveniles recently out of the nest also may be active during daylight hours (I. Montgomery, 2018, personal observation). The species has previously been reported to exhibit temporal variability in activity patterns between seasons (Miller & Elton, 1955; Wolton, 1983). Wood mouse detections were chance occurrences as none of the camera trap projects that comprise this study specifically focussed on small mammals. While small mammals may present identification challenges to camera trap studies (Claridge, Paull & Barry, 2010; Meek & Vernes, 2016), wood mice were attracted to arboreal baiting stations, often being the most recorded species. In contrast to other small mammals in Ireland, wood mice are known to be arboreal, so the methodology used could be applied elsewhere in the world to improve our knowledge of cryptic small mammals.

Anti-predator behaviours, which include avoidance (Curé et al., 2013), facilitate the survival of prey by mitigating predation (Sih & Christensen, 2001). In the current study, foxes consistently occurred in sympatry with hares and rabbits, both of which are putative prey species (Reynolds & Aebischer, 1991). Both predator-prey annual cross-correlations indicated that foxes and lagomorphs are likely to be active simultaneously. In temperate zones, foxes typically mate in late winter/early spring; litters may contain up to 12 cubs, with food availability being a significant factor (Larivière & Pasitschniak-Arts, 1996). Here we describe foxes as becoming increasingly crepuscular in late-spring and summer, thus increasing the potential for temporal overlap with both species of lagomorph. This suggests that predation of these species may increase during the fox breeding season. Indeed, lagomorphs may become an increasingly important food source as the cubs grow, particularly if the vixen has many offspring, as both lagomorphs are amongst the most substantial meals available to a medium-sized terrestrial predator in Northern Ireland. Given the relative lack of carnivores in the country, we can be reasonably confident that the behavioural repertoires of both hares and rabbits in Northern Ireland include fox-specific anti-predator behaviours. Pine marten and squirrels showed a direct negative correlation in the annual comparison, as well as during spring, summer, and winter. Thus, predator and prey activity are simultaneously different (i.e. when pine martens are most active, squirrels are least active and vice-versa). Squirrels are also subject to predation by diurnal birds of prey (Petty, Lurz & Rushton, 2003); by avoiding nocturnal predators, the animals become diurnally vulnerable. However, while squirrels have relatively poor nocturnal vision (Arden & Silver, 1962); their spatial acuity improves under brighter conditions (Jacobs, Birch & Blakeslee, 1982). Foraging during the night thus greatly increases the risk of predation.

It was not possible to optimise the present study a priori, comprising, as it does, several individually-designed surveys. For example, there was considerable variation in the effective densities of camera trapping arrays. Higher density arrays increase the chances of capturing an animal in-transit, being placed near a resting site (i.e. sett, drey, form, etc.), and capturing the focal species if it occurs at low densities. Camera trap surveys, therefore, would ideally consistently use high-density arrays to return an abundance of data. This is rarely feasible, however, given time, personnel, and financial constraints, all of which were limiting factors to the contributing surveys, though, the involvement of trained citizen scientists spreads workload and increases the amount or data available for analysis (Swanson et al., 2015). However, one of the key strengths of camera trap research is the ability to collate data from disparate, methodologically unique studies to describe fundamental ecological parameters. Moreover, the potential importance of bycatch data from camera trap surveys cannot be over-stated and it is important that data can be aggregated across independent surveys to answer relevant questions.

In contrast to terrestrial animals that move on a 2D plain, arboreal animals move within 3D space, thus decreasing the likelihood of a random capture in transit. Squirrels were enticed to specific locations for image capture by the use of baited stations. While baiting is certainly effective, it is not without its problems. For example, animals may identify bait stations as a reliable source of food, and thus frequently revisit them, thereby inflating counts (Rowcliffe & Carbone, 2008). Moreover, feeding animals may spend a considerable amount of time in front of the camera if undisturbed than those captured in transit, again affecting interpretations. The degree of uncertainty increases considerably where the focal species does not exhibit individually-identifiable colouration or markings. Captures from baited stations, therefore, may only represent one, or a handful of individuals (Trolle & Kéry, 2003; Weckel, Giuliano & Silver, 2006). However, the present study is only concerned with activity, and, hence, the detection of any individual during a given time period was assumed to be representative of the species as a whole.

It should also be noted that the data used in the present study are uncorrected for detection probability. Thus, observed patterns could represent changes in animal activity, abundance or detectability over time. For example, ambient noise can affect individual behaviour (Francis & Barber, 2013), subsequently impacting the number of detections recorded during camera trap surveys along with associated data and inferences. While we acknowledge that many factors will impact the detectability of a species, including the height of emergent vegetation, camera placement, lighting condition (e.g. daytime/night time) and the density and behaviour of the focal species (O’Brien, Kinnaird & Wibisono, 2003; Royle & Nichols, 2003; Silveira, Jácomo & Diniz-Filho, 2003; Larrucea et al., 2007; Harmsen et al., 2010; O’Connell, Nichols & Karanth, 2010; Chandler, Royle & King, 2011; Meek et al., 2014; Cusack et al., 2015; Meek, Ballard & Falzon, 2016), we are confident that, given our sample sizes, the data presented herein reflect actual activity patterns of the reported species.

Understanding the activity patterns of wildlife, and seasonal variations thereof, is of considerable benefit in furthering our understanding of species ecologies and informing future research (e.g. the development and application of efficient ecological surveys), thus paving the way for the development of appropriate management policies and/or conservation programmes. Knowing when a species is most or least likely to be active can lead to considerable methodological improvements, including potentially reducing the probability of achieving false-negatives, particularly for scarce or cryptic species. Camera trap surveys seeking to investigate circadian and intra-annual species activity patterns should, ideally, be conducted over the course of an entire year, focussing on areas which the focal species are known to frequent, and employing a large number of traps. Furthermore, additional data such as climate, topography and habitat may further inform interpretations and facilitate the application of statistical models. While the temporal distribution of data herein are arguably suboptimal due to the application of varied (i.e. non-standardised) methodologies, and environmental data are lacking, they are nevertheless of great utility in describing fundamental aspects of species’ ecologies.

Conclusions

Camera traps allow practitioners to concurrently survey across a wide range of species and habitats, providing data that may be of great utility in informing subsequent investigations and/or answering important ecological questions. In the present study, we draw together several disparate, and very different camera trap surveys to describe fundamental behavioural parameters of 10 mammal species. We found that focal-species-specific and associated bycatch data derived from camera traps are effective in providing insight into the daily lives of mammals. In particular, they have provided an effective means of describing circadian activity patterns and seasonal variations in temporal activity. In addition, they have utility in investigating temporal aspects of interspecific interactions and directing further research into such relationships. Invasive alien species are an ongoing issue and mitigating the threats posed to Irish hares and red squirrels by their respective conspecifics and potential predators requires accurate ecological data to effectively guide conservation efforts. Furthermore, prey switching is an issue with regards to carnivores of conservation interest (e.g. pine martens) and shifts in the activity patterns of disease vectors (e.g. deer, badgers) are relevant to their control. As many of the species described herein are of economic, management and/or conservation interest, our data will help inform the development of appropriate species-specific methodologies and processes and associated policies.

Supplemental Information

Supplemental Information 1 Figure S1. Activity patterns of European hares (Lepus europeaus) and Irish hares (L. timidus hibernicus).

Due to the close similarities in activity patterns, species were grouped as ‘hares’ for the purposes of this study.

Click here for additional data file.

Supplemental Information 2 Figure S2. Activity patterns of grey squirrels (Sciurus carolinensis) and red squirrels (Sciurus vulgaris) and Irish hares (L. timidus hibernicus).

Due to the close similarities in activity patterns, species were grouped as ‘squirrels’ for the purposes of this study.

Click here for additional data file.

Supplemental Information 3 Figure S3. Survey sites with detections of one of ten mammal species observed in camera trap studies in Northern Ireland.

(a) Badger (Meles meles), (b) fallow deer (Dama dama), (c) fox (Vulpes vulpes), (d) hare (Irish hare, Lepus timidus hibernicus and European hare, L. europaeus), (e) pine marten (Martes martes), (f) rabbit (Oryctolagus cuniculus), (g) squirrel (grey squirrel, Sciurus carolinensis and red squirrel, S. vulgaris), and (h) wood mouse (Apodemus sylvaticus). • = species detected at that location; ○ = species not detected at that location.

Click here for additional data file.

Supplemental Information 4 Figure S4. Overlap between diel activity patterns of (A) fox and hare, (B) fox and rabbit, (C) fox and wood mouse, (D) pine marten and squirrel, and (E) pine marten and wood mouse.

The overlap coefficient (Δ) ± 95% Confidence Intervals (CIs) are also given.

Click here for additional data file.

Thanks to the great many support staff, fieldworkers and landowners who were involved in the surveys which contributed to this manuscript. We also thank three reviewers, whose feedback helped guide improvements to the manuscript.

Additional Information and Declarations

Competing Interests

Author Contributions

Data Availability

The authors declare that they have no competing interests.

Anthony Caravaggi conceived and designed the experiments, performed the experiments, analysed the data, contributed reagents/materials/analysis tools, prepared figures and/or tables, authored or reviewed drafts of the paper, approved the final draft.

Maria Gatta analysed the data, prepared figures and/or tables, authored or reviewed drafts of the paper, approved the final draft.

Marie-Claire Vallely contributed reagents/materials/analysis tools, authored or reviewed drafts of the paper, approved the final draft.

Kayleigh Hogg performed the experiments, contributed reagents/materials/analysis tools, authored or reviewed drafts of the paper, approved the final draft.

Marianne Freeman performed the experiments, contributed reagents/materials/analysis tools, authored or reviewed drafts of the paper, approved the final draft.

Erfan Fadaei performed the experiments, contributed reagents/materials/analysis tools, authored or reviewed drafts of the paper, approved the final draft.

Jaimie T.A. Dick conceived and designed the experiments, contributed reagents/materials/analysis tools, authored or reviewed drafts of the paper, approved the final draft.

W. Ian Montgomery conceived and designed the experiments, contributed reagents/materials/analysis tools, authored or reviewed drafts of the paper, approved the final draft.

Neil Reid conceived and designed the experiments, contributed reagents/materials/analysis tools, authored or reviewed drafts of the paper, approved the final draft.

David G. Tosh conceived and designed the experiments, performed the experiments, contributed reagents/materials/analysis tools, authored or reviewed drafts of the paper, approved the final draft.

The following information was supplied regarding data availability:

GitHub: https://github.com/arcaravaggi/PeerJ_mammal_activity

Caravaggi A, Gatta M, Vallely M-C, Hogg K, Freeman M, Fadei E, Dick J, Montgomery WI, Reid N, Tosh D. 2018. arcaravaggi/PeerJ_mammal_activity: Code & data archive (Version v1.0.0). Zenodo. http://doi.org/10.5281/zenodo.1420397.

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
