# Peer review of "Seasonal and predator-prey effects on circadian activity of free-ranging mammals revealed by camera traps"

_PeerJ, doi:10.7717/peerj.5827_

## Round 0.1 · original submission · Major Revisions

You have received three very thorough reviews of your manuscript, and I would ask you to address the reviewers' comments point-by-point in an accompanying letter and highlight in the manuscript where these changes have been made.

Reviewer 1 ·

Basic reporting

The raw data is accessible and for the most part, the study contains good, intelligible English. A few typos and examples of poor wording are mentioned in the line-specific comments below.

In terms of scope and questions, I believe that the authors might be over-reaching with what inference they can draw on predator-prey relationships (especially given that many of these prey species likely have numerous predators whose behavior is not documented in this study), but the circadian rhythm reporting is valid. More emphasis up front could be given about the limitations of the data (as discussed more fully in my comments).

Experimental design

The authors are obviously making good use of data that has been collected previously, but I think the authors need to spend more time discussing the limitations of using these opportunistic data sets. One of my biggest issues with the paper is that the authors don't spend much (any?) time acknowledging that they are using indices that are uncorrected for detection probability and could represent either changes in animal activity or changes in animal abundance or chances in detectability through time. For example, trigger distance could be shorter at night or fewer animals could be seen at night because visibility is worse. Seasonally, vegetation could obscure cameras more in summer vs. winter. The authors suggest that seasonal changes in capture rates are due to changes in behavior, but perhaps for some of these short-lived species, they are actually picking up on changes in population size? While the authors may not want to explore correcting for detectability within the scope of what they are trying to accomplish here, acknowledging these issues and citing appropriate literature (starting with O'Brien et al. 2011 on RAIs, for one) I think would be incredibly important for this paper.

Also, for some of these species, the empirical opportunistic data is worse or perhaps more un-comparable than others. I'm thinking specifically of the baited squirrel traps and the issues of the cameras not being at a height to capture small rodents. Perhaps these two species could be excluded from the analysis?

I also think the authors could do a better job of stressing in the beginning that little has been done to study the Irish populations of these animals. The discussion reads like they are finding over and over again that their measurements match the literature, such that highlighting why what the authors have accomplished here is either new or what repeating these studies adds to the literature would be helpful (I think this is there, just not well-expressed).

Validity of the findings

Another issue I had with this paper is the amount of inference that the authors attempt to draw on predator-prey dynamics in these systems. I believe that the information they present on circadian rhythms is valid, but many drivers dictate activity patterns, including physiological constraints and the presence of other predators uncaptured in this study. This is a very correlative study and the authors make very strong statements about predator-prey dynamics that I simply can't see as being supported by what they present here.

Additional comments

This paper describes the circadian rhythm patterns of Irish woodland animals derived from opportunistic camera trap data from a variety of survey projects. The authors also attempt to draw inference on temporal niche partitioning between these species in a predator-prey context, although as their work is only correlative as does not consider other key factors dictating animal circadian rhythms, I believe that some of these conclusions should be toned down.

I found the discussion a little bit difficult to wade though, going species by species. It would have been easier to grasp if the authors first could have highlighted any surprising or unusual findings first, and then talked about each prey in light of all of their predators at the same time (e.g., something like “woodmice temporal activity patterns overlapped heavily with pine martins, but they appeared to exhibit asynchrony with (which may indicate avoidance of) this other predator") to help facilitate the conclusions that the authors draw about predator-prey dynamics (although see my note above).

Line-by-line comments are presented below:

Line 29: And physiological constraints and evolutionary history and a whole host of other, critical drivers. I think physiological constraints (as per Kronfeld-Schor & Dayan 2003) at the very least need to be considered here.

Line 30: Remove "of deer, hare, and squirrel" because it confuses the reader in the next two lines about your actual study species

Line 47: First sentence is grandiose and not necessarily - delete.

Line 48: Do not need to define "circadian" to this audience

Line 49 - Adaptive behavior, yes, but also constrained by organism physiology (i.e., Kronfeld-Schor & Dayan 2003).

Line 52: For predator/prey activity, include Kronfeld-Schor & Dayan 2003, Fenn & McDonald 1994 at the very least.

Lines 50-54: This whole section could be bolstered by a few more references.

Lines 55-61: Just including a single animal as an example here is a little confusing and subjective, and not particularly useful. Either change to groups of animals that display these behaviors, or rather, just remove entirely.

Line 62: Also include Fenn & McDonald 1994 here.

Line 65: I don't know if this sentence follows from the previous thought. I would delete or move to a new paragraph

Lines 68-69: This would be a good place to say WHY we would want to do this in the first place.

Line 72: You mention a "number of methodological techniques", but then only list one - could you elaborate further by including other techniques and their flaws?

Lines 68-87: I would restructure this paragraph so that the methods are listed first, with the limitations of the methods at the end.

Line 90: There are a number of recent camera trap reviews which have come out in the last few years which would be more germane here, such as Wearn & Glover-Kapfer 2017.

Line 114: "Few focus on quantifying animal activity patterns" – I don’t think this is true at all. I would estimate that 70-80% of camera trapping papers quantify animal activity patterns.

Line 118: "hypothesize"; also, please visit Kronfeld-Schor & Dayan 2003 for how plastic animals can be in terms of changing their activity patterns - degree to which patterns are response of behavioral plasticity vs. evolutionary hardwiring / other constraints.

Line 118: What do you hypothesize predators will do in response to changes in activity by prey? I think it's important to remember that you're capturing the end play of a predator-prey game, where each player has been adjusting their own activity over evolutionary time in response to the other.

Line 123: Was there any way in which you quantified human activity? What was the gradient of human activity across your sites?

Line 142: Set randomly? Systematically? Stratified? Average distance between camera traps? How does this relate to the average home range of a deer (i.e., how likely is it that a single individual gets captured at multiple camera traps?) – same comments for all camera trap studies

Line 146: How many photos per trigger for each kind of camera specifically? Also, how many cameras of each type?

Line 148: Relocated? The authors stress in introduction that cameras can be deployed for an indefinite period of time…

Line 148-149: What did the authors do to correct for these differences in survey design?

Line 153: Typo: "either" not "either3"

Line 154/155: "in situ" not italicized here, but italicized above on line 151 – please be consistent.

Line 146: The authors don't list total number of locations for previous survey -- please be consistent in information presented about each survey (a comparative table would be helpful).

Line 166: In the methods, it would be good to discuss how difference in camera placement, like angle, sensitivity, etc., are dealt with statistically.

Line 170: How did the authors determine independent captures? 10 min delay, 30 min delay, etc.?

Line 174: How does this compare to other survey locations?

Line 178, 181: Average deployment?

Line 189: Do the authors have a citation for this? Perhaps look at the relative abundance index literature (e.g., O'Brien et al. 2003).

Line 195: These first two sentences can perhaps be less dramatic.

Line 198: This sentence should be rewritten without the weird clause (bad grammar).

Line 252: To what degree do the authors think seasonal differences are due to vegetation obscuring camera trap visibility (or similar)?

Line 318: Are these other values also in accordance with the literature? In which case, I would want an argument about what repeating the study adds to our understanding of these animals' behaviors.

Lines 326-328: Is there any reason to believe that fewer captures actually a result of lower population size, rather than chance in animal behavior? Is there a hunting season or similar for any of these species?

Line 332-334: Confusing grammar – change to something like: "It has been suggested that fallow deer are sensitive to anthropogenic disturbance, exhibiting facultative behavioral responses such as ..."

Line 348: This statement is too strong - the empirical study that you've just conducted with opportunistic data does not prove this statement.

Line 414: Type - "focused"

Line 429-430: This is too strong a statement to make about these animals’ ecology based on this data.

Reviewer 2 ·

Basic reporting

This paper is an interesting and well-written study of mammal camera trap data in Northern Ireland. The writing style is, on the whole, clear and unambiguous. I have a minor comments on basic reporting few comments, listed below.
The discussion and context are well referenced, and the figures and tables are informative and well-designed.
In the references in lines 55 to 61, it is not clear which reference is the recent study of 4,477 species and which are specific references describing the behaviour of individual species. This could be expressed more clearly.
Lines 121-122 – this statement about the climate is ambiguous – and doesn’t really need a Met Office reference to say that the climate is temperate and that precipitation varies locally. If you refer to the climate, at least give some statistics (annual rainfall, temperatures range etc…).
Line 127 – what is meant by “non-inclusive” here?
Figure 3 label “fallow” should be “fallow deer”.
Figure 5 column headings “annual”, “spring” etc… above these plot would be easier to interpret than the complicated labelling system.
Figure 5 and 6 – no need to state in the figure legend “Data were derived from camera trap surveys in Northern Ireland….” This is explained in the methods.
Line 308 “distinct between seasons” – do you mean consistent between seasons?

Experimental design

The experimental design is well defined, and although the disparate trap data utilised were not designed primarily to address the questions posed by the paper, the authors do a good job of showing that camera trap data can provide an effective method of describing activity patterns in mammals, and present some useful results on activity patterns in Irish mammals.

Validity of the findings

The findings are robust, and statistics are appropriate. The methods used to assess predator avoidance are most open to criticism, as correlations between activity in predators and prey does not constitute a causal mechanism (any diurnal or nocturnal species will show a correlation between any other diurnal or nocturnal species). However, the authors discuss their results appropriately, the conclusions are well stated, and do not overstate the significance of their findings.

Additional comments

Overall I found this a straightforward and interesting study, using an admirably comprehensive dataset to assess the use of camera traps in diurnal activity. Given PeerJ’s data policy, I would prefer if the data (say, hourly activity counts throughout the survey period for each species) were made available.

·

Basic reporting

I commend the authors for a well written article. The language is clear and unambiguous, professional. The authors have also adequately cited relevant literature to the research. Figures and tables have been provided. However, some of these maybe more suited as supplementary material, and for some the quality (resolution) will need improvement before final publication. I have also noticed that the same information (some results) is presented both in table and figure. I have suggested that results only be presented in one form and not both, preferably in table format. For instance, Figure 5 and Table 4 are presenting the same information. I thank the authors for sharing the raw data. The data itself is robust and worth a full publication testing the presented hypothesis.

Experimental design

The manuscript presents original research and fits within the aims and scope of PeerJ. The research question is well defined and relevant, filling an important knowledge gap. The authors have performed rigorous and technical analysis for a well thought through research question. However, the study designs from which the data is drawn are not adequately presented. The statistical analysis and its associated results are based on data drawn from several studies, with different designs and data outcomes, which has raised problems in the interpretation and should be improved before Acceptance. I suggest that the study would have bypassed this caveat by using data from the same population (s) collected with same methods and design. The inclusion of small mammals in camera trap data analysis also raises further questions on the validity of the interpretations considering that much of the camera trap literature suggests camera traps as ineffective means to survey small mammals. The authors may need to provide further justification for the inclusion of small mammals in their analysis.

Validity of the findings

The study is of high importance to wildlife ecology and specifically predator-prey interactions. Nevertheless, analyzing data from various sources causes lots of problems in interpretation for reasons the authors have explained. The camera trap analysis explained in this manuscript would have benefited a more streamlined study involving standardized data from one population or system. Further, data on small mammals brings more problems considering that camera traps are not the most effective methods to study small mammals. Indeed, most studies, even those investigating predator-prey relationships, rarely include small mammals in the analysis. I find the conclusions of the study to be of a narrow scope and do not specifically speak to the data, but general statements from camera trap studies. For instance, the last sentence of the abstract appears in virtually all camera trap studies on predator-prey interactions. Give one example how your data facilitate effective management and conservation of the ten species under study? What is novel about this data/study that readers have not read in existing camera trap literature? This is currently lacking and/or insufficiently presented.

Additional comments

General comments (some already highlighted in the three sections above) :
I commend the authors for for their extensive analysis of camera trap data combining several studies and species. The manuscript has been clearly written in a highly professional style. However, the statistical analysis and its associated results are based on data drawn from several studies, with different designs and data outcomes, which has raised problems in the interpretation and should be improved upon before Acceptance. I suggest that the study would have bypassed this caveat by using data from the same population (s) collected with same methods and design. The inclusion of small mammals in camera trap data analysis also raises questions on the validity of the interpretations considering that several camera trap literature suggests camera traps as ineffective means to survey small mammals. Circadian activity was not given much attention by the authors as compared to seasonal and annual patterns. I also suggest that the temporal activity categories should be confirmed with statistical tests. Please include legend on figure 1. For Fig. 2. I don't think images of the mammals is necessary. If necessary, I suggest images for all the 10 species are presented. For Fig. 3. I suggest that the y-axis title be changed to reflect that it represents activity density. Fig.6. Please advise what the dotted lines mean? Table 2. I suggest a column on body size is included in the table. Table 4. (title) I would rephrase " Temporal activity interactions (avoidance and attraction) of five predator-prey pairs" or something along those lines…. attraction?? Predators synchronizing their activity with that of their prey. I suggest that decimal places are consistent

Specific comments:
L2: I suggest “change” be replaced with “differences”. “Change” may suggest a causal effect which a correlative study like this one may not provide evidence for.
L2: Insertion: … temporal…
L22: Word count missing
L28: I suggest “occurrence” be replaced with “presence”
L29-32: Long sentence communicating different ideas. I suggest this sentence is split.
L30: The surveys were for deer, hare and squirrel, but the results you report starting at L32 include other species. These other species were presumably detected during the course of the three surveys. Thus, it may seem inaccurate to report surveys as "deer, hare and squirrel" surveys, when the surveys recorded other species.
L30: … to describe circadian and seasonal temporal activity of ten…
L31: Camera traps have been criticized as ineffective tools to studying small mammals. Please justify your use of camera traps to study small mammals. With this limitation, how representative are your species of interest of the small mammal community in your system?
L31: What about prey attraction behavior? Predator-prey interactions involve both prey avoiding predators and predators being attracted to prey.
L31: Helpful to describe this analysis to the reader, otherwise please cut
L32: Please rephrase. "We recorded 8,761 detections”. Please also mention if the detections were independent or not.
L32-L35: I have five thoughts on how this result is presented:
1) I don't think to report the n if we even don't know whether these were independent detections or not.
2) The species should be presented in some order may be based on body size or number of detections, the latter being in ascending or descending order.
3) The way the circadian activity is reported is not consistent. Some species were reported as "largely".... while for others the pattern is specifically reported e.g. as crepuscular or diurnal.
4) Are the reported circadian patterns based on a statistical test i.e. are the badgers significantly nocturnal? If yes, would be useful to report these.
5) The study species should be clearly mentioned when results are reported. This is particularly true for the two species of hares. Please give the names for both species. The same applies to "rabbits".
L36: Sentence may need to be rephrased. Currently hard to follow.
L39: … cut “highly”
L40: I suggest If there is a need to categorize some of the species as lagomorphs, please do early enough in the text. For instance, which of the 10 species in your study are being referred to as lagomorphs?
L40: Is this a comparison among seasons? How many seasons were considered for the analysis? This should be mentioned somewhere above.
L41: Is this a comparison among seasons? How many seasons were considered for the analysis? This should be mentioned somewhere above.
L41: The reader will not be surprised by this statement. The ability of camera trap studies to provide ecological information on a wide range of species (excluding small and non-terrestrial mammals) is pretty established by a wealth of published camera trap studies.
L43: Please cut “which may improve…”
L43-44: I find the conclusions of the study to be of a narrow scope and do not specifically speak to the data, but general statements from camera trap studies. For instance, the last part of the sentence appears in virtually all camera trap studies on predator-prey interactions. Give one example how your data facilitate effective management and conservation of the ten species under study? What is novel about your data/study that readers have not read in existing camera trap literature? This is currently lacking and/or insufficient.
L47: Spatially too
L48: Spatial patterns too... I suggest that would be really nice to have this paragraph speaking to both spatial and temporal activity as two key niche processes.
L50: This sentence here resonates with my thoughts that this first paragraph should be rephrased to reflect the contribution of spatial activity too. The way it is presented here it ignores the important contribution of spatial activity to species ecology. Then in the second paragraph, I would narrow down to temporal activity, and speak to both circadian and seasonal temporal activity.
L55: I suggest that this paragraph is rephrased to introduce in detail literature on temporal activity. I like the last sentence of this paragraph. I would build on such to make a case why temporal activity is an important niche process in animals.
L55: Worth reminding the reader that "activity patterns" is specifically speaking to temporal activity
L56: I would cut this text. I don't see how it is contributing to the story. Further, the examples of nocturnal, diurnal, cathemeral or crepuscular species given here are very species specific, and makes me wonder why you chose these species as examples out of the 4477 species you have mentioned at L56. Perhaps only mentioning the four main temporal activity categories (and their definitions) without going into specific examples will suffice.
L61: This is an interesting citation, but I don't its relevance to the subject matter of the manuscript. The title speaks to "free-ranging mammals". Yet, text here speaks to laboratory subjects. Further, your findings do not show any intraspecific variation in temporal activity. I suggest you cut.
L67: Please define “season”
L71: I suggest you cut. The first sentence of this paragraph excellently highlights the challenges of studying animal temporal activity. The next sentence should highlight how camera traps have become a standard means to filling this knowledge gap.
L88: Please give a brief definition of what these are.
L90: Please be specific by directly mentioning these "population parameters"
L94: Please cut “survey effort and”
L98: Cut text
L100: Already mentioned at L94. Please cut.
L101: I would cut text from L101 to L112. Firstly, this text undermines your use of camera traps to study wildlife activity. Secondly, it contradicts some of the benefits of camera trap data you have highlighted in the preceding text. What I would find useful is to have this paragraph end with a justification for using camera traps to study the wildlife in your system, particularly small mammals, for which, camera traps have been reported to be ineffective to study.
L114: Please give examples
L115: Insertion “… temporal… activity patterns (circadian and seasonal)”
L117: I don't see how your data is investigating predation risk. What your data is showing is avoidance (for prey) or attraction (for predators), which may reflect predator-prey interactions e.g. predation risk.
L118: Precisely! This is what (temporal avoidance) the study is investigating, not predation risk. However, I would add to this hypothesis that predators will be attracted to prey by being active when their prey is active. It is an arms-race -between prey avoiding predators and predators getting attracted/synchronized to prey.
L119: I suggest that the method section is re-written. Differences in study design including number of camera placement, camera trap models, baiting vs. no baiting, positioning from the ground, number of photos per trigger, duration of trigger delay and number of survey days, both within and between species need clarification. Further, the study designs for the included studies should be explained in detail than currently presented. There is also repetition of text in the methods e.g. "cameras were equipped with fitted with 8 GB HDSD cards,156 secured with Python security cables, motion detectors were set to medium sensitivity, each capture was stamped with the date and time" appears in methods for each species (or group). The methods section needs to be streamlined to avoid such repetition.
L120: Active tone please. I would rephrase "We collected camera trap data on deer, squirrels...."
L121: Is this a general description of weather conditions at the study site or description during the study period? My guess is the former, as I don't think climate has changed since 2016.
L122: Same thoughts as on L121. I guess the tense should be tweaked to reflect this is a description of the study site and not for the study period.
L123: Please cut “individual”
L124: Please rephrase to active tone
L125: Please give species or genus names
L126: Please give species or genus names
L127: Please give species or genus names
L127: Please explain what non-inclusive means.
L128: Please cut “current”
L134: When were the badgers, foxes, rabbits and wood mouse surveyed? They don't appear in the survey schedules outlined in the preceding sentence. Were these recorded as part of the deer, squirrel and hare surveys?
L134: This is a result. Please move to results. Please also include the test statistic for this result.
L135: Please highlight earlier in the Introduction that activity patterns refers to the circadian and seasonal activity.
L140: Please define acronym DS
L140: please rephrase to active tone
L141: I suggest “squares” be replaced with “grids”
L142: The numbers do not add up. It is also not clear what the sampling design was. Please give the number of cameras (and camera placements) used for each of the four and two deer studies, and not averages. How many cameras (and camera placements) per 1 sq-km grid?
L148: Again, why the difference in design for the same species?
L149: Another difference in design; 30 vs. 40 cm from the ground, number of triggers, delay... Please explain the difference.
L151: The whole study was in-situ. Please clarify. I don't see the need for mentioning in situ. Please just say " Cameras were left at locations for 7 days". This applies to most of the methods succeeding this section.
L151: Why 7 and not 14? like for other DS?
L151: Please explain the difference in the design among the DS. There is variation in design ranging from camera trap models to placement and survey period. These differences need to be explained.
L157: Please justify the 1-hour threshold and give references to previous studies.
L161: 23 1km2 grids
L168: Videos for triggers
L168: Cut in-situ
L168: Does this mean that detection differs between conspecifics? How does the difference in still image vs. video influence detection of conspecifics?
L169: This a big assumption (and an important part) of the study design. Justification of its use should be included here and not referenced.
L176: 3-4 m from the ground for a small mammal? This seems too high?
L177: The deer and hare studies did not involve baiting. Why the difference? Was the baiting targeting only the squirrels and martens or also the predators? Baiting vs. no baiting makes a huge difference in number of detections and may even bias temporal activity. Please see recent work documenting the impacts of baiting in camera trap studies.
L181: Differences in design
L182: Again, I think this is way too high from the ground for mammals as small as squirrels. This needs to be justified.
L188: Please explain the logic behind the assumption. Animals can only be detected on camera traps only when active, specifically, when in motion. The number of detections per diel hour period reflects the level of activity across the circadian period.
L189: What was the rationale behind the grouping? Was it for the purposes of identifying independent events?
L190: This sentence is unclear. Please rephrase. Mentioning that photos recorded an hour apart were assumed independent detections is intuitive enough.
L192: Inserted texted “in a single camera photo”
L205: Please “cut”
L207: What was the independent and dependent variable s for the ANOVAs. Please provide this information here.
L209: active tone please?
L210: What about at the circadian scale?
L211: Which is a predator and prey? Please show in parenthesis
L212: Cut text
L213: Which is a predator and prey? Please show in parenthesis
L219: Active tone please.
L227: Are all these independent detections? If yes, please mention. What was the total survey effort?
L227: Please present in order e.g., descending order. I also find the camera trap images in Fig.2 for the mammals redundant.
L235: It would be great to have these patterns tested statistically i.e., is the number of species detections (activity) statistically different across the circadian periods (nocturnal vs. diurnal vs. cathemeral vs. crepuscular)?
I also suggest that the circadian and seasonal patterns are presented separately; starting with the circadian patterns followed by the seasonal patterns. Same applies to results on the overlap coefficients and lagged regressions.
Figure 5 and Table 4 are presenting the same information. I suggest that results are presented in one or the other, but not both. I suggest table.
I suggest that the analysis also include temporal activity at the circadian scale, in addition to annual and seasonal.
L239: Text to start on new paragraph
L244: Text to start on new paragraph
L268: The text in this section is mixed. I suggest that overlap coefficients (and their CIs) be presented separately from the lagged regressions.
L269: What about at a circadian scale?
L270: Were the overlap estimates generated from the overlap package? If yes, they should be referred to Fig. 3.
L270: Figure number is missing
L272: Table 4 and Fig. 5 show the same information. I suggest that results only presented in either table or figure but not both. I suggest the table format.
L273: This sentence should go to the discussion
L274: I would not consider a 75% overlap between predator and prey a "temporal separation". This depicts attraction or synchrony. I would rephrase "... of temporal overlap..."
L274: Again, I don't think this result reflects predation risk. I would rephrase "... suggesting increased opportunities for prey (hare) encounter (hunting success) for the fox..." or something along those lines.
L280: Which species? Please clarify by mentioning the species
L283: Is this last sentence of this paragraph an overall summary for all species pairs or for a specific predator-prey pair? Its not clear. If yes to the latter, perhaps, the pred-prey pair should be mentioned somewhere in this sentence.
L303: I feel that the circadian activity patterns have not been given much focus in the study, but seasonal and annual.
L304: I don't think "change" is the best word to use here. "Change" may imply a causal effect of sunrise/sunset on temporal activity patterns, which may require a controlled experiment and not a correlative study like this one. I would rephrase "... exhibited different activity patterns in relation to sunrise/sunset" or something along those lines.
L305: Please see my suggestion at L304
L307: Cut “months”
L309: Insert “consistent”
L309: This contradicts the earlier suggestions that attribute differences in temporal activity due to sunrise/sunset and season
L310: Please cut “particularly”
L311: Insert “… activity…”
L312: May you please define here and in the methods what overlap coefficient is considered "low or high"?
L313: winter
L313: Long sentence. Please split text on marten and squirrels
L318: Please check author guidelines. Chronological??
L320: between
L321: … reported… limited
L324: I suggest you cut. The discussion should focus on the activity patterns.
L325: … less active…
L326: Move reference to the end of the sentence.
L328: This is critical. And you highlighting it here undermines your findings, and indeed underpins the weakness of the study design. Maybe the animals were not just detected because some of the studies had cameras positioned way high above the ground???
L329: Please cut “found to be”..
L330: cut text
L331: cut text
L332: I wish you had statistical tests to confirm these. "Largely" and similar words are subjective. Please see my earlier thoughts on this earlier in the text.
L334: Please move to L333 after "... behavioural responses..."
L337: cut text “found to be…”
L337: "Increased" suggests a change, and possibly, a causal effect. I suggest the tone is tweaked to reflect that this a correlative study and not experimental. That said, I would rephrase to "... nocturnal; being more active at dusk and less at dawn".
L340: Cut text “rural”
L343: Timing at which scale; circadian, seasonal or annual? please clarify
L345: Lethal control is not a disturbance. A disturbance is a general stimulus, and animals may get habituated or sensitized to disturbances (see Frid and Dill 2002). On the other hand, lethal control is a specific stimulus (shooting in this case), and animals would be expected to respond with specific anti-predator strategies, otherwise animals stand high risk of mortality (predation). Please rephrase.
L348: Patterns for “periods”
L348: Using this word, and not earlier in the text, may be confusing. Further, it is best to be consistent with the animal names throughout the text.
L350: Please cut percentages in parenthesis.
L353: … than in…
L354: May you explain why this may be the case? Could it be body size???
L358: This sentence seems redundant and doesn't add much to the story. I will cut.
L360: It is possible that the species can co-exist via spatial or dietary niche partitioning.... or even avoid each other temporally at very fine scale. There is a great deal of literature, much of from sympatric carnivore systems.
L363: This sentence is unclear. ... a suggestion of underlying...? Who is making the suggestion?
L365: This sentence is hard to follow without the reader's prior knowledge of Monterroso et al., (2013) findings.
L366: You raise a good point here! Indeed, the use of different datasets from various studies causes lots of problems in interpretation for reasons you explain well here. I suggest that the study would have bypassed that caveat by using data from the same population collected with same methods and design. The current analysis leaves open myriad explanations for the observed patterns than what is reported here.
L371: “Sporadic” should be explained.
L373: cut text
L375: This is irrelevant to your results. Please cut
L378: Inserted text “associated with its”
L379: Cut text “of contemporary studies”
L382: But for “bar”
L383: …higher…
L388: cut text
L390: Inserted text “between seasons”
L392: Please give reference here
L395: Please specify the populations being referred to here
L395: I don't see relevance. I suggest you cut.
L404: Is 73 independent events of a free-ranging animal few?
L408: reported to
L412: The categorizations of "small" or "low" are really subjective. For instance, an ecologist working with free living wildlife would perceive 57 a large sample size. I suggest rephrase.
L413: Again, this statement undermines your work. Much of your prey data is for small mammals. Further, as I have already mentioned, camera traps are perhaps not the best means to survey small mammals.
L417: This is not what the extensive camera trap literature shows.
L418: See my preceding thoughts on using data collected with designs for varying purposes for this analysis.
L422: … cut text “direct”…
L423: Cut text
L430: ?
L434: This study did not test spatiotemporal relationships but temporal.
L463: as I have mentioned in my thoughts throughout this text.
L475: Baiting will definitely heighten animal activity, and the implications of this should definitely be a factor in this analysis. There is a possibility that animals will alter their activity patterns to exploit increased forage resource.
L490: The current analysis and its results are hard to interpret for reasons you have well highlighted. The camera trap analysis explained in this ms would have benefited a more streamlined study involving standardized data from one population or system. Further, data on small mammals brings more problems considering that camera traps are not the most effective methods to study small mammals. Indeed, most studies, even those investigating predator-prey r/ships, rarely include small mammals in the analysis.
L500: This is open to debate, as clearly pointed out in your discussion. I would suggest that study is better streamlined to focus on medium-to-large system in survey(s) that employed standardized methods, and in so doing mitigating study design related confounding factors.
L501: Please rephrase. Currently complicated to comprehend.

---

## Round 0.2 · accepted · Accept

I believe that you and your coauthors have done an excellent job in addressing the detailed and thorough comments from the three reviewers. I do not believe there is a need to redo your statistical analysis as suggested by reviewer 3. The revised manuscript is very much improved with a much clearer message.

#